# Role of Inflammation in Canine Primary Glaucoma

**DOI:** 10.3390/ani14010110

**Published:** 2023-12-28

**Authors:** Lionel Sebbag, Oren Pe’er

**Affiliations:** Koret School of Veterinary Medicine, The Hebrew University of Jerusalem, Rehovot 76100, Israel

**Keywords:** anti-inflammatory medications, glaucoma, goniodysgenesis, intraocular pressure, ocular inflammation, pigment dispersion, trabecular meshwork, uveitis

## Abstract

**Simple Summary:**

Glaucoma is a common and devastating eye disease in dogs, resulting in pain and blindness if not treated promptly and appropriately. Primary glaucoma develops when the pressure inside the eye (IOP) is elevated in the absence of other diseases inside the eye; it is the most common form of glaucoma in dogs and is reported in many canine breeds (genetic predisposition). The main trigger for glaucoma is IOP elevation, but there are other important contributing factors, such as inflammation inside the eye. Here, we summarize all the evidence that links inflammation to primary glaucoma in dogs. A better understanding will allow clinicians to better manage primary glaucoma in dogs and scientists to examine newer therapeutic targets for improved clinical outcomes.

**Abstract:**

Primary glaucoma is a painful, progressive, and blinding disease reported in many canine breeds, characterized by intraocular pressure (IOP) elevation in the absence of antecedent intraocular disease. Clinical observations of dogs with primary glaucoma suggest that many affected eyes develop concurrent intraocular inflammation in addition to elevated IOP. In this work, we summarize the current knowledge that relates inflammation to primary glaucoma in dogs, reviewing studies focused on genetics, physiology, histopathology, bioanalysis of ocular fluids, therapeutics, and clinical outcomes of glaucomatous patients. Through disruption of the blood–aqueous and blood–retinal barriers, pigment dispersion, and biochemical changes to the aqueous humor and tear film, the pathogenesis of canine primary glaucoma appears to involve inflammatory changes to various extents and with various consequences from the front to the back of the eye. Among others, inflammation further impacts IOP by reducing aqueous humor outflow at the level of the iridocorneal angle and accelerates vision loss by promoting neuronal degeneration. As such, the vicious cycle of ocular inflammation and IOP elevation might warrant the use of anti-inflammatory medications as a core component of the treatment regime for dogs with primary glaucoma, either therapeutically (i.e., actively glaucomatous eye) or prophylactically in the yet unaffected contralateral eye.

## 1. Introduction

Canine glaucoma—a painful, progressive, and blinding condition associated with intraocular pressure (IOP) elevation and subsequent optic nerve/retinal atrophy—can be broadly classified as congenital, primary, or secondary [1]. Primary glaucoma is characterized by IOP elevation in the absence of antecedent intraocular disease and is reported in many canine breeds where genetic predisposition is documented or presumed. Primary glaucoma is subdivided into open-angle (POAG) and angle-closure (PACG) types. Dogs affected by POAG initially have open iridocorneal angles in the early disease stages until the ciliary cleft collapses with end-stage glaucoma [1]. Given its usefulness as a model for human POAG, there are numerous publications describing the clinical manifestations, structural changes, inheritance, genetics, and pathophysiology of POAG in beagle dogs [1]; however, POAG remains very rare as a spontaneous form of primary glaucoma in canine patients to date [1]. In contrast, canine PACG is related to pectinate ligament dysplasia (also known as goniodysgenesis) and represents the most common form of primary glaucoma in canine patients [1,2]. Clinical observations of dogs with primary glaucoma suggest that many affected eyes develop concurrent intraocular inflammation in addition to elevated IOP [1]; the inflammation can be subtle (i.e., low-grade/subclinical uveitis) or quite pronounced, notably during episodes of acute congestive glaucoma in certain breeds such as the Basset Hound [3] and the American Cocker Spaniel [4]. Regardless of the intensity, intraocular inflammation can negatively affect the pathogenesis and prognosis of primary glaucoma, for instance, by reducing aqueous humor outflow (e.g., fibrin/inflammatory cells deposits in the trabecular meshwork, peripheral anterior synechiae) or by promoting neuronal degeneration [1,5,6].

A recent survey showed that most practitioners (61.2%) never use anti-inflammatory medications as part of prophylactic therapy in canines with primary glaucoma [7]. When the response was affirmative (38.8%), some clinicians mentioned they only use anti-inflammatory drugs when there is clinical evidence of inflammation on examination (e.g., aqueous flare, hypotony) [7]. Another survey of UK veterinarians regarding acute glaucoma management showed that topical corticosteroids were considered by 9.1% (general practitioners) and 20% (ophthalmology graduates) of respondents, while topical nonsteroidal anti-inflammatory drugs (NSAIDs) were considered by 3.6% (general practitioners) and 0% (ophthalmology graduates) of the respondent [8]. In the authors’ opinion, such infrequent or inconsistent use may be due (at least partly) to a misconception that ocular inflammation is only associated with secondary glaucoma in dogs [1,9] and/or misinterpretation that only noticeable inflammation can be detrimental to the pathogenesis of glaucoma. In humans, the relationship between inflammation and primary glaucoma is well described, with a comprehensive review highlighting glaucoma-related inflammatory processes in all ocular structures, from the back to the front of the eye and beyond [10].

Although the veterinary literature on this topic is more limited, there is mounting evidence that inflammation also plays an important role in the pathophysiology of primary glaucoma in the canine species. In this work, we summarize the current knowledge that relates inflammation to primary glaucoma in dogs, reviewing studies focused on genetics, physiology, histopathology, bioanalysis of ocular fluids, therapeutics, and clinical outcomes of glaucomatous patients, specifying whether the findings apply to canine POAG and/or PACG where relevant.

## 2. Pathophysiology of Canine Glaucoma and Inflammation

### 2.1. Compromised Blood–Aqueous Barrier

Canine eyes with primary glaucoma have compromised vascular integrity within the anterior uvea as a consequence of elevated IOP [1,11]. When compared to healthy controls, iris angiography of canine eyes with POAG had significantly prolonged times to onset of arterial, capillary, and venous phases, as well as slower onset but greater severity in fluorescein leakage within the aqueous humor [11]. Such delayed perfusion and disruption to the blood–aqueous barrier may indicate pressure-associated anterior-segment ischemia (ASI) in dogs with primary glaucoma, as mentioned in a recent review by Drs. Miller and Bentley [1]. The same pathophysiology is true in human patients with primary glaucoma, in whom the higher the IOP elevation, the greater the mechanical pressure on the blood–aqueous barrier, the more susceptible the ASI, and the more severe the intraocular inflammation [12].

Pressure-associated ASI and breakdown of the blood–aqueous barrier can result in several changes that can be observed in canine eyes with acute congestive glaucoma, including:***Aqueous flare and aqueous cell***, which may be more severe with the amplitude of IOP elevation (as shown in human patients with POAG) [12] and may worsen after a dramatic reduction in IOP as previously hypoxic tissues become reperfused [1].***Dilated pupil*** that is fixed or sluggish in response to light stimulus, either related to atrophy of the iris (sectorial or diffuse) and associated iris constrictor muscle [1], afferent pupillary defect, or other mechanisms that are poorly described.***Dispersion of pigment*** released by iris atrophy and chafing of the posterior iris epithelium against the lens, causing pigmentary dusting on the iris surface, anterior lens capsule, corneal endothelium, and iridocorneal angle [1,5,13].***Alterations in aqueous humor biochemical composition***, discussed in more detail in Section 2.3.

In chronically affected POAG or PACG eyes, intraocular inflammation can result in peripheral anterior synechiae [11,13,14], preiridal fibrovascular membranes [1,13,15,16], and potential damage to trabecular meshwork cells [16,17].

### 2.2. Compromised Blood–Retinal Barrier

A study of canine eyes with acute (≤5 days from clinical signs) or chronic (>5 days) PACG showed glaucoma-related disruption of retinal pigmented epithelium (RPE), increased permeability of the retinal vascular endothelium, pigmentary dispersion, accumulation of inflammatory cells within the neuroretina and vitreous body, as well as accumulation of serum albumin around retinal blood vessels [18]. In another study of acute PACG (≤5 days from clinical signs), early histopathological changes were noted in the retina and optic nerve of canines with primary glaucoma, including neutrophilic and histiocytic inflammation at different stages of the disease [19]. Together, these mechanisms may expose numerous epitopes during the rapid neuronal cell death, resulting in the formation of autoantibodies directed against retinal antigens [6]. In turn, T cell-mediated immune response may accelerate retinal and optic nerve degeneration that was initiated by glaucoma, independent of IOP elevation [6,20].

### 2.3. Biochemical Changes in Ocular Fluids

A recent study by Terhaar and colleagues measured 15 pro-inflammatory cytokines in the aqueous humor of dogs with either early or end-stage PACG, postoperative ocular hypertension (POH) following phacoemulsification, or no ocular pathologies (healthy control) [21]. Primary glaucomatous eyes had significantly higher levels of tumor necrosis factor alpha (TNF-α) and interleukin 18 (IL-18) when compared to the POH and control eyes, and IL-18 levels were positively correlated with increasing IOP, findings that may indicate inflammation plays a role in the pathogenesis of primary glaucoma in dogs [21]. In American Cocker Spaniel dogs with end-stage PACG (i.e., prior to ciliary body chemical ablation), a proteomic analysis of the aqueous humor showed significantly higher protein concentration in eyes with primary glaucoma vs. healthy eyes, with pathways related to inflammation being significantly upregulated in glaucomatous eyes [4]. Specifically, secreted phosphoprotein 1 (SPP1), a protein that is expressed by different types of immune cells (e.g., neutrophils, macrophages, and lymphocytes) and trabecular meshwork tissues [22], was 81-fold higher in glaucomatous vs. healthy eyes [4]. Aqueous humor levels of other biomarkers are also elevated in canine eyes with primary glaucoma [23], including TNF-α [24], matrix metalloproteinases (MMP-2, MMP-9) [25], and transforming growth factor-beta (TGF-β). Interestingly, the upregulation of inflammatory pathways can also be observed in the tear fluid of dogs with primary glaucoma (POAG and PACG), with differential expression levels of proteins from glaucomatous vs. healthy eyes, including pro-inflammatory cytokines (Table 1) [26]; of note, most glaucoma cases in that study were described as chronic (>1 month duration) although some cases were in the early stages of the disease (<1 week duration).

Of note, similar biochemical changes were reported in the tears and aqueous humor of human patients with POAG [27], findings that could serve as prognostic factors for disease outcome [28].

### 2.4. Histopathological Changes

A morphologic review was performed by Reilly et al. (2005) on 100 canine eyes enucleated due to end-stage PACG. In that study, pigmentary dispersion was noted in 96% of all examined globes, along with posterior iris pigmented epithelium loss or stripping [5]. Further, neutrophilic inflammation was noted in 64.7% of acute cases and 16.7% of chronic cases, suggesting an association between inflammatory cells and congestive glaucoma in dogs; however, it remains unclear if the ocular inflammation noted in that study represented a trigger for glaucoma or if it was secondary to pigment dispersion or degenerative changes in the iridocorneal angle [5]. Histological changes related to inflammation were also described in other reports of canines affected by PACG, including peripheral anterior synechiae and neutrophilic/histiocytic inflammation in the retina and optic nerve [14,19].

### 2.5. Genetic Markers of Inflammation

A recent study assessed the genetics of Basset Hound dogs with normal iridocorneal angles (controls), pectinate ligament abnormalities, or PACG [29]. Using genome-wide association studies (GWASs), most of the genes that statistically differed among groups had functions related to inflammation and immunity [29]. In another study of canine eyes with end-stage PACG, the authors identified over 500 genes with statistically significant changes in expression levels between the glaucomatous and healthy canine retinas, with most genes being associated with inflammation such as antigen presentation, protein degradation, and innate immunity [6].

### 2.6. Ocular Surface and Systemic Inflammation

***Ocular surface inflammation***—The detrimental impact of antiglaucoma medications on the ocular surface has been extensively reported in the human literature, with evidence of inflammation and disrupted ocular surface homeostasis from the majority of antiglaucoma drug classes, including carbonic anhydrase inhibitors, prostaglandin analogs, β-blockers, α-adrenergic, cholinergic, and ROCK inhibitors [10,30]. In humans, inflammatory/pathological changes to the ocular surface are diverse and include meibomian gland dysfunction, tear film deficiency, blepharitis, keratopathies (e.g., superficial punctate keratitis, reduced corneal sensitivity), and various conjunctival pathologies (e.g., hyperemia, keratinization, fibrosis, squamous metaplasia, and goblet cell deficiency) [30]. In contrast, very little is known about this topic in veterinary patients. Keratitis, conjunctivitis, and blepharitis have been reported in dogs as an adverse effect of topical carbonic anhydrase inhibitors [31,32], while keratoconjunctivitis and other ocular surface pathologies (e.g., corneal hypoesthesia and neurotrophic ulcers) were commonly observed in glaucomatous canine eyes undergoing transscleral cyclophotocoagulation [33,34].

***Systemic inflammation***—In humans, complete blood counts of patients with primary glaucoma showed elevated platelet-to-lymphocyte ratio (PLR) and neutrophil-to-lymphocyte ratio (NLR) when compared to healthy controls, a finding that could indicate a potential role of systemic inflammation in the pathogenesis of glaucoma [35,36]. In fact, a high PLR represents a risk factor for glaucoma progression in humans, being associated with more rapid visual field loss and a higher likelihood of developing progressive retinal ganglion cell damage [37]. In dogs, a study by Boillot et al. showed no differences in plasma levels of inflammatory markers (i.e., C-reactive protein, haptoglobin, and serum albumin) between controls and dogs predisposed to primary glaucoma, although the study did not assess systemic inflammation in dogs with active glaucoma [38].

## 3. A Self-Perpetuating Vicious Cycle of Inflammation and Glaucoma

On the ocular surface, glaucoma-related inflammation can have serious implications in terms of prognosis and disease progression. Inflammation can directly affect medication compliance, for instance, owing to drug discontinuation due to local adverse effects. Inflammation can also reduce drug bioavailability due to blood-tear barrier breakdown and subsequent protein binding in the tear film [39,40,41,42]. Further, irritation from chronic glaucoma medications and associated toxic preservatives (e.g., benzalkonium chloride) can diffuse from the surface to deeper structures, leading to degeneration and inflammation of the trabecular meshwork and higher resistance to aqueous humor outflow [30,43]. This vicious cycle might clinically appear as an apparent loss of medication efficacy, namely, the more drugs, the greater the toxic reaction, leading to a higher IOP despite the initial therapeutic response [43].

Inside the eye, glaucoma-related inflammation negatively impacts the health of the trabecular meshwork, neurosensory retina, and optic nerve, further exacerbating IOP elevation and vision loss. Trabecular meshwork degeneration and remodeling occur due to the aforementioned pigment dispersion (i.e., antigenic stimulation in response to melanin and/or other mechanisms) [44], as well as pro-inflammatory cytokines in the aqueous humor of glaucomatous eyes. For instance, TGF-β is well known to promote fibrosis through extracellular matrix remodeling in the trabecular meshwork, further reducing aqueous humor drainage [45]. Further, the inflammation, disruption of the RPE, and increased vascular permeability (i.e., blood–retinal barrier breakdown) may potentiate oxidative stress to the retina [46], expose retinal autoantigens to the immune system [6,19,47] and thereby initiate or contribute to neuronal damage.

The vicious cycle of inflammation and glaucoma is also relevant in canine patients undergoing filtering surgeries with gonioimplants [48]. Growth of the fibrotic capsule forming around the gonioimplant (bleb fibrosis), ultimately responsible for implant failure and IOP elevation, is believed to be stimulated (at least in part) by the ‘toxic’ inflamed aqueous humor of glaucomatous dogs that is drained in the subconjunctival space around the gonioimplant [48,49,50]. Moreover, antiglaucoma medications contribute to the development of subconjunctival fibrosis [30], further reducing the drainage of aqueous humor via the filtering device. As such, limiting inflammation appears to be a major factor in the control of bleb fibrosis [48,49,50].

## 4. Anti-Inflammatory Therapy for Eyes Predisposed or Affected by Primary Glaucoma

### 4.1. Impact of Anti-Inflammatory Drugs on IOP

***Corticosteroids***—Corticosteroid-induced IOP elevation is relatively common in humans, presumably related to lowered aqueous humor outflow from the deposition of extracellular material in the trabecular meshwork [51]. In contrast, most corticosteroids do not appear to influence IOP in dogs, including hydrocortisone [52], prednisone [53], prednisolone acetate [54], triamcinolone acetonide [55], and difluprednate [56]. For triamcinolone acetonide, intravitreal injection of the long-acting corticosteroid did not induce sustained IOP elevation over 3 months of monitoring IOPs [55]. For difluprednate, wiltype purpose-bred beagle dogs received topical difluprednate 2–3 times daily for an extended period (413–764 days) as controls in a gene therapy trial; selected dogs developed periocular alopecia and suppression of the hypothalamic-pituitary-adrenal axis, however none were reported to have ocular hypertension as an adverse effect of chronic corticotherapy (Dr. András Komáromy, personal communication) [56]. The one exception appears to be dexamethasone, with IOP shown to increase following a single intravenous administration of dexamethasone in healthy dogs [57], and following repeated topical 0.1% dexamethasone in beagles with inherited POAG [58]. In the study by Gelatt and Mackay (1998), the elevation in IOP was relatively mild (~4.8 mmHg) and would likely be lower in nonglaucomatous canine eyes; further, the mild IOP elevation rapidly resolved following cessation of dexamethasone administration [58].

***Non-steroidal anti-inflammatory drugs (NSAIDs)***—Orally administered carprofen or meloxicam did not influence IOP in normal dogs [53,59]. However, a study of topical flurbiprofen showed IOP elevation in dog eyes [60], presumably due to greater resistance in aqueous humor outflow [61]; in fact, immediate postoperative use of topical flurbiprofen may be a potential risk factor for the development of glaucoma following phacoemulsification in dogs [62].

***Concurrent anti-inflammatory and prostaglandin analogs***—NSAIDs and corticosteroids inhibit the formation of prostaglandin from arachidonic acid via the inhibition of cyclooxygenase or phospholipase A, respectively [63]; consequently, these drugs could theoretically inhibit the indirect action of prostaglandin analogs, that is, synthesis of endogenous prostaglandins that further act on prostaglandin F2α (FP) receptors. In dogs, concurrent administration of latanoprost and flurbiprofen [60], or latanoprost and prednisolone acetate [64], resulted in a notable reduction of the ocular hypotensive effect relative to latanoprost therapy alone (20.4% and 56%, respectively) [60,64]. Interestingly, a delay between eye drops administration may negate the aforementioned impact of anti-inflammatory drugs; in fact, Kahane et al. showed that the ocular hypotensive effect of PGF-2α analog was relatively similar in canine eyes receiving latanoprost alone vs. latanoprost then prednisolone acetate 3 h later [54].

### 4.2. Anti-Inflammatory Therapy in Glaucomatous Dogs

***Prophylactic***—Prophylactic therapy is critical for the yet unaffected eye in canine patients with primary glaucoma (Table 2) [7,32,65,66,67,68,69,70].

A prospective multicenter clinical trial showed that demecarium bromide along with a topical corticosteroid (betamethasone), both applied once daily, equally delayed the onset of glaucoma in the unaffected eye when compared to canine eyes receiving twice daily betaxolol (median 31 months and 30.7 months, respectively) [65]. As such, the authors speculated that the combination of antiglaucoma + anti-inflammatory was preferred owing to the less frequent dosing schedule, a key component for long-term compliance. In a retrospective study by Dees and colleagues (2014), the estimated median time to medical failure (IOP ≥ 20 mmHg) for canine eyes receiving topical antiglaucoma and anti-inflammatory medication was 324 days versus 195 days in eyes receiving antiglaucoma medication alone [66]. Although not statistically significant, this important finding was deemed ‘clinically significant’ as glaucoma onset was delayed by over 66% in eyes receiving concurrent anti-inflammatory medication [66].

***Therapeutic***—In dogs, the use of anti-inflammatory therapy for patients with PACG has been reported in book chapters and review articles [1,2,71], albeit not critically evaluated to date. In humans, a recent randomized controlled trial evaluated the management of acute PACG using antiglaucoma therapy with (steroid group) or without (control) a single subconjunctival injection of dexamethasone [72]. Twenty-four hours after initial treatment, the steroid group had significantly lower IOP, lower severity of conjunctival hyperemia, lower ciliary flush, and pain when compared to the control group; further, the success rate at 24-h (i.e., IOP between 6–21 mmHg) was significantly better in the steroid vs. control group (79.9% vs. 54.9%, respectively) [72]. In another study of glaucomatous eyes undergoing a filtering surgery, the 5-years outcome was similar whether using a local antifibrotic agent (mitomycin C) or long-acting corticosteroid (triamcinolone acetonide), emphasizing the potential benefits of corticosteroids (or other drugs with antifibrotic effects) following glaucoma surgery [73]. Similar findings may be true in dogs with primary glaucoma, especially given species similarities in the inflammation profile related to primary glaucoma [10,21,23,74], although this speculation should be verified in future prospective studies.

### 4.3. Miscellaneous

It is known that primary glaucoma requires long-term therapy and that corticosteroids are systemically absorbed following topical administration in dogs [75,76]. As such, for patients considered at risk (e.g., low body weight, pre-existing endocrinopathy), clinicians might select alternatives to prednisolone acetate, such as loteprednol etabonate, a ‘soft’ corticosteroid with a lower potential for systemic adverse effects [77]. Further, glaucoma-related inflammation could be targeted more precisely than broad anti-inflammatory therapy, for instance, TNF-α reduction via competitive inhibition of TNF receptors (e.g., etanercept) in ocular tissues [78]. Last, nutraceutical compounds could also assist in reducing inflammatory-related influence on glaucoma progression and prognosis [10]. For instance, the flavonoid myricetin was shown to lower inflammatory cytokines in the aqueous humor and trabecular meshwork [79], while omega-3 fatty acid supplementation reduced pro-inflammatory cytokines in tears and improved dry eye symptoms in human patients with glaucoma [80,81]. In veterinary medicine, two recent surveys showed that nutraceutical use is relatively common for glaucomatous dogs [7,82]; however, well-designed prospective studies are desperately needed as today’s evidence is still uncertain and inconclusive [83].

## 5. Conclusions

Through disruption of the blood–aqueous and blood–retinal barriers, pigment dispersion, and other mechanisms, the pathogenesis of canine primary glaucoma appears to involve inflammatory changes to various extents and with various consequences from the front to the back of the eye. It remains unclear, however, whether inflammation is a trigger or mostly a bystander of primary glaucoma in canine eyes; in fact, several of the canine studies reported in this review involved canine eyes with chronic or end-stage primary glaucoma, where inflammatory changes may actually represent a sequela of chronically elevated IOPs. Regardless, the vicious cycle of intraocular inflammation and IOP elevation might warrant the use of anti-inflammatory medications as a core component of the treatment regime for dogs with primary glaucoma, either therapeutically (i.e., actively glaucomatous eye) or prophylactically in the yet unaffected contralateral eye.

## Figures and Tables

**Table 1 animals-14-00110-t001:** Pro-inflammatory cytokines and other proteins identified at higher levels in ocular fluids of canine eyes with primary glaucoma compared to healthy eyes.

Ocular Fluid	Disease Characteristics	Compounds	Reference
Aqueous humor (cytokines)	Early and end-stage primary angle-closure glaucoma, various breeds	IL-4, IL-8, IL-18, TNF-α	[21]
Aqueous humor (proteomics)	End-stage primary angle-closure glaucoma, American Cocker Spaniels	SPP-1, PGLYRP2, YWHAE, CPN1, MGAM, APOC2, VIM, LOC476104, PON1, CCL23, CrybA3, CPN2, CHI3L1, ANGPTL3, MMP19, APOE, LOC100684663, PIGR, MFAP4, AHSG, TF, APOA1, CCL14, GPLD1, C9, SERPINC1, ITIH2, ACTB, F12, CD14, SMPDL3A, AGT, C3, C4BPA, CLU, IGFALS, AFM, C1QC, ACTR3B, ACTR3C, C2, TIMP1, HABP2, F2, FETUB, F9, F10, PEPD, SERPINF2, HPX, GC, VTN, C7, SHBG, APOD, ITIH1, KLKB1, HGFAC, CFI, CTSC, FN1, LCP1, ALB	[4]
Tear film (proteomics)	Early and chronic primary glaucoma (open-angle and angle-closure), various breeds	Complement C3, Cytosol aminopeptidase, Cytosolic non-specific dipeptidase, Debrin-like protein, GTP-binding nuclear protein Ran, Poly(rC)-binding protein 1, Heterogenous nuclear ribonucleoprotein, Histine triad nucleotide-binding protein, Nicotinate phosphoribosyl-transferase, Osteoclast-stimulating factor 1, Protein ABHD14B, Protein-L-isoaspartate(D-aspartate) O-methyltransferase, Tumor protein D52, V-type proton ATPase catalytic subunit A, Apolipoprotein, Transforming protein RhoA, Nucleoside disphosphate kinase, Retinol binding protein, NSFL1 cofactor p47, Thymosin beta-4, Nucleoside disphosphate kinase, WD repeat-containing protein 1	[26]

**Table 2 animals-14-00110-t002:** Median time to glaucoma onset in the second eye following diagnosis of primary angle-closure glaucoma (PACG) in the first eye, in the absence of prophylactic therapy, or in the presence of antiglaucoma medications with or without anti-inflammatory medications.

Authors (Year)	No Therapy	Antiglaucoma	Antiglaucoma + Anti-inflammatory	Reference
Slater and Herb, 1986	5 months	10 months	–	[69]
Miller et al., 2000	8 months	30.7 months	31 months	[65]
Strom et al., 2011	–	19.2 months	–	[70]
Park et al., 2012		14 months (mean 17.5 months)		[68]
Dees et al., 2014	–	195 days	324 days	[66]
Stavinohova et al., 2015	–	9.2 months	–	[32]
Ahn et al., 2022	–	20.3 months	–	[67]

## Data Availability

No new data were created or analyzed in this study. Data sharing is not applicable to this article.

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
