# Peer review of "Role of Inflammation in Canine Primary Glaucoma"

_animals, 2023, doi:10.3390/ani14010110_

Round 1

Reviewer 1 Report

Comments and Suggestions for Authors

The authors provide an excellent, comprehensive review of the role of inflammation in canine primary glaucoma. Clearly, inflammation is involved in this disease, and the authors point out correctly that detailed mechanisms are unknown. All of my comments are minor concerns and suggestions to improve the manuscript:

- The authors should emphasize which cited studies were performed on endstage glaucomatous eyes. Most studies documenting the presence of inflammation in primary glaucoma were completed late in the disease process, with therapies complicating interpretation, suggesting that inflammation may be secondary rather than primary.

- The authors should be more specific throughout the text if referring to PACG or POAG. Most of the cited canine studies were done in PACG.

-  This manuscript provides an excellent opportunity to put steroid-induced ocular hypertension in perspective. As far as I know, there are no documented studies showing that the use of topical steroids facilitates the formation of glaucoma. This is different in humans, where steroid-induced glaucoma is a genuine concern. Even in Beagles with ADAMTS10-open-angle glaucoma, topical dexamethasone only elevates IOP in the single-digit range (mmHg).

-  In line 92, the authors mention mydriasis is mainly the result of iris atrophy. However, increased IOP without atrophy will also result in mydriasis, for example, when the anterior chamber of a cadaver eye is inflated with water, salt solution, or viscoelastics. In other words, it is complicated to determine how much atrophy plays a role.

- Table 1: It would be helpful to include the breed and the type of glaucoma since the changes may not apply to all forms of primary canine glaucoma. For example, the study by Yun et al. (2021) was performed in American Cocker Spaniels with endstage glaucoma.

- Lines 164 & 165: myocilin may not be a biomarker of canine glaucoma. I think it is elevated secondary to IOP increases. The third mutated gene identified as a cause for primary glaucoma, in addition to ADAMTS10 and ADAMTS17, is OLFML3 in Border Collies with PACG. In general, the names of genes should be written in italics.

-  Line 247: The title “Anti-inflammatory therapy in glaucomatous dogs” should be numbered 4.2.

-  Table 2: Please specify that this refers to PACG, not POAG.

Author Response

Thank you for the constructive feedback! See attached PDF.

Reviewer 2 Report

Comments and Suggestions for Authors The authors present a good summary of the associations between canine primary glaucoma and inflammatory cells, cytokines, and other pathophysiologic changes associated with inflammation. The paper is overall written well with good organization and presentation of pertinent data from the current literature. The authors present some interesting data and discussion regarding inflammation and glaucoma. Although I do not reject the authors' premise that there is an association, there are a few errors and misleading statements that must be addressed prior to publication, and I think that more data and discussion is needed in order to justify publishing a review on this topic.  

In the introduction, the authors describe different types of primary glaucoma and variations in pathogenesis; however, the authors need to be more thorough and explicitly state whether each inflammatory factor they discuss is contributing to primary open angle glaucoma, primary closed angle glaucoma, or both.

  The authors often refer to pigment dispersion as an inflammatory process (e.g., line 57). The current literature considers pigment dispersion to be a degenerative change that occurs due to age, anatomical variations of the iris, etc.; it is not considered an inflammatory process. The study they later cite on this topic (Reilly et al., 2005) clearly describes pigment dispersion as a histopathological finding separate from inflammation.   I recommend removal of the first sentence of section 2.2 (line 103). This sentence implies to the reader that all of the following changes to the eye are due to inflammation, but that is not true. Some of the changes the authors list are associated with inflammation but are not directly caused by inflammation, and some changes can be seen in eyes without any histopathological inflammatory changes.   In line 147, the percentages are incorrect. The authors use 86% and 50%, which are the percentages for <4 days and between 4-7 days, respectively. Neutrophils were present in 64.7% of acute cases and in 16.7% of chronic cases in that study. In line 148, the phrase "suggesting that inflammation plays a key factor in the pathogenesis of acute congestive glaucoma" is a little misleading, since the presence of inflammatory cells is not a definitive indicator of an active inflammatory process, and the remainder of the paragraph correctly discusses how the inflammation may be a secondary or incidental finding rather than an integral component of the pathogenesis. The phrase "suggesting an association between inflammatory cells and congestive glaucoma" is more accurate wording.   In line 164, ADAMTS10 has indeed been associated with inflammation in the published literature, listed clearly by the NIH: https://www.ncbi.nlm.nih.gov/gene/81794.   In general, I would like to see Section 4 expanded on if possible, since the effects of anti-inflammatory drugs would be the most compelling evidence of inflammation having a role in causing or progressing primary glaucoma rather than being an incidental finding. Table 2 shows only two studies that looked at the effects of anti-inflammatory drugs, and the differences compared to anti-glaucoma drugs alone seem underwhelming. From the description in section 4.1, it sounds like the elevation of IOP by corticosteroids would seem to exacerbate glaucoma rather than prevent or treat it, so the authors need to further discuss the clinical usefulness of these drugs for these cases. In line 276-277, the authors state that corticosteroids are important for preventing fibrosis within the eye, but that is not necessarily an anti-inflammatory application. The decreased fibrosis may be due to the suppressive effects of corticosteroids on fibroblast activation rather than by preventing inflammation.

Comments on the Quality of English Language

The quality of English is overall very good. There are a few minor grammatical/typographical edits that are recommended:

- Remove the comma after "dogs" on line 10 - Add a comma after "elevation" on line 11 - Remove the comma after "disease" on line 39 - The sentence from line 125-127 should be slightly rearranged for reading clarity, and I recommend this phrasing: "Specifically, secreted phosphoprotein 1 (SPP1), a protein that is expressed by different types of immune cells (e.g., neutrophils, macrophages, lymphocytes) and trabecular meshwork tissues [22], was 81-fold higher in glaucomatous vs. healthy eyes [4]." - Line 128: should read "levels of other biomarkers are" (not "is"). - Line 247: this should be section 4.2 (section 4.1 was already listed on line 224).

Author Response

Thank you for the constructive feedback! See attached PDF

Round 2

Reviewer 2 Report

Comments and Suggestions for Authors

The authors have significantly strengthened their paper with these edits and the addition of new details. I believe this paper is scientifically sound and contributes to the veterinary medical literature by offering a thorough and meaningful review of inflammation and canine glaucoma, and no further edits are necessary.